# Automatically Discovering and Learning New Visual Categories with Ranking Statistics

**Kai Han**[*]     **Sylvestre-Alvise Rebuffi**[*]     **Sebastien Ehrhardt**[*]
**Andrea Vedaldi**     **Andrew Zisserman**
Visual Geometry Group, Department of Engineering Science, University of Oxford
`{khan,srebuffi,hyenal,vedaldi,az}@robots.ox.ac.uk`

## Abstract

We tackle the problem of discovering novel classes in an image collection given labelled examples of other classes. This setting is similar to semi-supervised learning, but significantly harder because there are no labelled examples for the new classes. The challenge, then, is to leverage the information contained in the labelled images in order to learn a general-purpose clustering model and use the latter to identify the new classes in the unlabelled data. In this work we address this problem by combining three ideas: (1) we suggest that the common approach of bootstrapping an image representation using the labeled data only introduces an unwanted bias, and that this can be avoided by using self-supervised learning to train the representation from scratch on the union of labelled and unlabelled data; (2) we use rank statistics to transfer the model's knowledge of the labelled classes to the problem of clustering the unlabelled images; and, (3) we train the data representation by optimizing a joint objective function on the labelled and unlabelled subsets of the data, improving both the supervised classification of the labelled data, and the clustering of the unlabelled data. We evaluate our approach on standard classification benchmarks and outperform current methods for novel category discovery by a significant margin.

## 1 Introduction

Modern machine learning systems can match or surpass human-level performance in tasks such as image classification (Deng et al., 2009), but at the cost of collecting large quantities of annotated training data. Semi-supervised learning (SSL) (Oliver et al., 2018) can alleviate this issue by mixing labelled with unlabelled data, which is usually much cheaper to obtain. However, these methods still require some annotations for each of the classes that one wishes to learn. We argue this is not always possible in real applications. For instance, consider the task of recognizing products in supermarkets. Thousands of new products are introduced in stores every week, and it would be very expensive to annotate them all. However, new products do not differ drastically from the existing ones, so it should be possible to discover them automatically as they arise in the data. Unfortunately, machines are still unable to effectively learn new classes without manual annotations.

In this paper, we thus consider the problem of discovering new visual classes automatically, assuming that a certain number of classes are already known by the model (Hsu et al., 2018; 2019; Han et al., 2019). This knowledge comes in the form of a *labelled dataset* of images for a certain set of classes. Given that this data is labelled, off-the-shelf supervised learning techniques can be used to train a very effective classifier for the known classes, particularly if Convolutional Neural Networks (CNNs) are employed. However, this does not mean that the learned features are useful as a representation of the *new classes*. Furthermore, even if the representation transfers well, one still has the problem of identifying the new classes in an unlabelled dataset, which is a clustering problem.

We tackle these problems by introducing a novel approach that combines three key ideas (section 2 and fig. 1). The first idea is to pre-train the image representation (a CNN) using all available images, both labelled and unlabelled, using a self-supervised learning objective. Crucially, this objective *does not* leverage the known labels, resulting in features that are much less biased towards the labelled

---

[*]indicates equal contribution

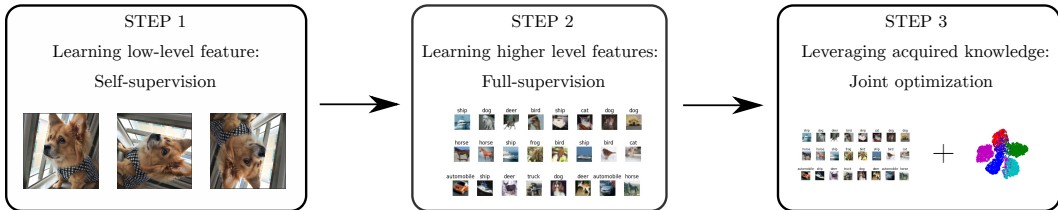

Figure 1: **Overview of our three step learning pipeline**. The first step of the training consists in learning an unbiased image representation via self-supervision using both labelled and unlabelled data, which learns well the early layers of the representation; in the second step, we fine-tune only the last few layers of the model using supervision on the labelled set; finally, the fine-tuned representation is used, via rank statistics, to induce clusters in the unlabelled data, while maintaining a good representation on the labelled set.

classes. Labels are used only after pre-training to learn a classifier specific to the labelled data as well as to fine-tune the deepest layers of the CNN, for which self-supervision is not as effective.

The second idea is a new approach to transfer the information contained in the labelled images to the problem of clustering the unlabelled ones. Information is transferred by sharing the same representation between labelled and unlabelled images, motivated by the fact that the new classes are often similar to the known ones. In more detail, pairs of unlabelled images are compared via their representation vectors. The comparison is done using robust rank statistics, by testing if two images share the same subset of $k$ maximally activated representation components. This test is used to decide if two unlabelled images belong to the same (new) class or not, generating a set of noisy pairwise pseudo-labels. The pseudo-labels are then used to learn a similarity function for the unlabelled images.

The third and final idea is, after bootstrapping the representation, to optimise the model by minimizing a joint objective function, containing terms for both the labelled and unlabelled subsets, using respectively the given labels and the generated pseudo-labels, thus avoiding the forgetting issue that may arise with a sequential approach. A further boost is obtained by incorporating incremental learning of the discovered classes in the classification task, which allows information to flow between the labelled and unlabelled images.

We evaluate our method on several public benchmarks (section 3), outperforming by a large margin all existing techniques (section 4) that can be applied to this problem, demonstrating the effectiveness of our approach. We conclude the paper by summarizing our findings (section 5). Our code can be found at http://www.robots.ox.ac.uk/~vgg/research/auto_novel.

## 2 METHOD

Given an *unlabelled* dataset $D^u = \{x_i^u, i = 1, \dots, M\}$ of images $x_i^u \in \mathbb{R}^{3 \times H \times W}$, our goal is to automatically cluster the images into a number of classes $C^u$, which we assume to be known *a priori*. We also assume to have a second *labelled* image dataset $D^l = \{(x_i^l, y_i^l), i = 1, \dots, N\}$ where $y_i^l \in \{1, \dots, C^l\}$ is the class label for image $x_i^l$. We also assume that the set of $C^l$ labelled classes is disjoint from the set of $C^u$ unlabelled ones. While the statistics of $D^l$ and $D^u$ thus differ, we hypothesize that a general notion of what constitutes a "good class" can be extracted from $D^l$ and that the latter can be used to better cluster $D^u$.

We approach the problem by learning an image representation $\Phi : x \mapsto \Phi(x) \in \mathbb{R}^d$ in the form of a CNN. The goal of the representation is to help to recognize the known classes and to discover the new ones. In order to learn this representation, we combine three ideas, detailed in the next three sections.

### 2.1 SELF-SUPERVISED LEARNING

Given that we have a certain number of labelled images $D^l$ at our disposal, the obvious idea is to use these labels to bootstrap the representation $\Phi$ by minimizing a standard supervised objective such as the cross-entropy loss. However, experiments show that this causes the representation to overly-specialize for the classes in $D^l$, providing a poor representation of the new classes in $D^u$.

Thus we resist the temptation of using the labels right away and use instead a self-supervised learning method to bootstrap the representation $\Phi$. Self-supervised learning has been shown (Kolesnikov et al., 2019; Gidaris et al., 2018) to produce robust low-level features, especially for the first few layers of typical CNNs. It has the benefit that no data annotations are needed, and thus it can be applied to both labelled and unlabelled images during training. In this way, we achieve the key benefit of ensuring that the representation is initialized without being biased towards the labelled data.

In detail, we first pre-train our model $\Phi$ with self-supervision on the union of $D^l$ and $D^u$ (ignoring all labels). We use the RotNet (Gidaris et al., 2018) approach[1] due to its simplicity and efficacy, but any self-supervised method could be used instead. We then extend the pre-trained network $\Phi$ with a classification head $\eta^l : \mathbb{R}^d \to \mathbb{R}^{C^l}$ implemented as a single linear layer followed by a softmax layer. The function $\eta^l \circ \Phi$ is fine-tuned on the labelled dataset $D^l$ in order to learn a classifier for the $C^l$ known classes, this time using the labels $y_i$ and optimizing the standard cross-entropy (CE) loss:

$$L_{\text{CE}} = -\frac{1}{N} \sum_{i=1}^{N} \log \eta^l_{y_i}(z^l_i) \tag{1}$$

where $z^l_i = \Phi(x^l_i) \in \mathbb{R}^d$ is the representation of image $x^l_i$. Only $\eta^l$ and the last macro-block of $\Phi$ (section 3) are updated in order to avoid overfitting the representation to the labelled data.

## 2.2 Transfer learning via rank statistics

Once the representation $\Phi$ and the classifier $\eta^l$ have been trained, we are ready to look for the new classes in $D^u$. Since the classes in $D^u$ are unknown, we represent them by defining a relation among pairs of unlabelled images $(x^u_i, x^u_j)$. The idea is that similar images should belong to the same (new) class, which we denote by the symbol $s_{ij} = 1$, while dissimilar ones should not, which we denote by $s_{ij} = 0$. The problem is then to obtain the labels $s_{ij}$.

Our assumption is that the new classes will have some degree of visual similarity with the known ones. Hence, the learned representation should be applicable to old and new classes equally well. As a consequence, we expect the descriptors $z^u_i = \Phi(x^u_i)$ and $z^u_j = \Phi(x^u_j)$ of two images $x^u_i, x^u_j$ from the new classes to be close if they are from the same (new) class, and to be distinct otherwise.

Rather than comparing vectors $z^u_i, z^u_j$ directly (e.g., by a scalar product), however, we use a more robust rank statistics. Specifically, we rank the values in vector $z^u_i$ by magnitude. Then, if the rankings obtained for two unlabelled images $x^u_i$ and $x^u_j$ are the same, they are very likely to belong to the same (new) class, so we set $s_{ij} = 1$. Otherwise, we set $s_{ij} = 0$. In practice, it is too strict to require the two rankings to be identical if the dimension of $z^u_i$ is high (otherwise we may end up with $s_{ij} = 0$ for all pairs $(i, j), i \neq j$). Therefore, we relax this requirement by only testing if the *sets* of the top-$k$ ranked dimensions are the same (we use $k = 5$ in our experiments), i.e.:

$$s_{ij} = \mathbb{1}\left\{\text{top}_k(\Phi(x^u_i)) = \text{top}_k(\Phi(x^u_j))\right\}, \tag{2}$$

where $\text{top}_k : \mathbb{R}^d \to \mathcal{P}(\{1, \ldots, d\})$ associates to a vector $z$ the subset of indices $\{1, \ldots, d\}$ of its top-$k$ elements.

Once the labels $s_{ij}$ have been obtained, we use them as pseudo-labels to train a comparison function for the unlabelled data. In order to do this, we apply a new head $\eta^u : \mathbb{R}^d \to \mathbb{R}^{C^u}$ to the image representation $z^u_i = \Phi(x^u_i)$ to extract a new descriptor vector $\eta^u(z^u_i)$ optimized for the unlabelled data. As in section 2.1, the head is composed of a linear layer followed by a softmax. Then, the inner product $\eta^u(z^u_i)^\top \eta^u(z^u_j)$ is used as a score for whether images $x^u_i$ and $x^u_j$ belong to the same class or not. Note that $\eta^u(z^u_i)$ is a normalized vector due to the softmax layer in $\eta^u$. This descriptor is trained by optimizing the *binary cross-entropy* (BCE) loss:

$$L_{\text{BCE}} = -\frac{1}{M^2} \sum_{i=1}^{M} \sum_{j=1}^{M} [s_{ij} \log \eta^u(z^u_i)^\top \eta^u(z^u_j) + (1 - s_{ij}) \log(1 - \eta^u(z^u_i)^\top \eta^u(z^u_j))]. \tag{3}$$

---

[1] We present to the network $\Phi$ randomly-rotated versions $Rx$ of each image and task it with predicting $R$. The problem is formulated as a 4-way classification of the rotation angle, with angle in $\{0°, 90°, 180°, 270°\}$. The model $\eta \circ \Phi(Rx)$ is terminated by a single linear layer $\eta$ with 4 outputs each scoring an hypothesis. The parameters of $\eta$ and $\Phi$ are optimized by minimizing the cross-entropy loss on the rotation prediction.

Furthermore, we structure $\eta^u$ in a particular manner: We set its output dimension to be equal to the number of new classes $C^u$. In this manner, we can use the index of the maximum element of each vector $\hat{y}_i^u = \text{argmax}_y[\eta^u \circ \Phi(x_i^u)]_y$ as prediction $\hat{y}_i^u$ for the class of image $x_i^u$ (as opposed to assigning labels via a clustering method such as $k$-means).

## 2.3 JOINT TRAINING ON LABELLED AND UNLABELLED DATA

We now have two losses that involve the representation $\Phi$: the CE loss $L_{\text{CE}}$ for the labelled data $D^l$ and the pairwise BCE loss $L_{\text{BCE}}$ for the unlabelled data $D^u$. They both share the same image embedding $\Phi$. This embedding can be trained sequentially, first on the labelled data, and then on the unlabelled data using the pseudo-labels obtained above. However, in this way the model will very likely forget the knowledge learned from the labelled data, which is known as *catastrophic forgetting* in incremental learning (Rebuffi et al., 2017; Lopez-Paz & Ranzato, 2017; Shmelkov et al., 2017; Aljundi et al., 2018).

In contrast, we jointly fine-tune our model using both losses at the same time. Note that most of the model $\Phi$ is frozen; we only fine-tune the last macro-block of $\Phi$ together with the two heads $\eta^u$ and $\eta^l$. Importantly, as we fine-tune the model, the labels $s_{ij}$ are changing at every epoch as the embedding $\eta^l$ is updated. This in turn affects the rank statistics used to determine the labels $s_{ij}$ as explained in section 2.2. This leads to a "moving target" phenomenon that can introduce some instability in learning the model. This potential issue is addressed in the next section.

## 2.4 ENFORCING PREDICTIONS TO BE CONSISTENT

In addition to the CE and BCE losses, we also introduce a consistency regularization term, which is used for both labelled and unlabelled data. In semi-supervised learning (Oliver et al., 2018; Tarvainen & Valpola, 2017; Laine & Aila, 2017), the idea of consistency is that the class predictions on an image $x$ and on a randomly-transformed counterpart $tx$ (for example an image rotation) should be the same. In our case, as will be shown in the experiments, consistency is very important to obtain good performance. One reason is that, as noted above, the pairwise pseudo-labels for the unlabelled data are subject to change on the fly during training. Indeed, for an image $x_i^u$ and a randomly-transformed counterpart $tx_i^u$, if we do not enforce consistency, we can have $\text{top}_k(\Phi(x_i^u)) \neq \text{top}_k(\Phi(tx_i^u))$. According to eq. (2) defining $s_{ij}$, it could result in different $s_{ij}$ for $x_i^u$ depending on the data augmentation applied to the image. This variability of the ranking labels for a given pair could then confuse the training of the embedding.

Following the common practice in semi-supervised learning, we use the *Mean Squared Error* (MSE) as the consistency cost. This is given by:

$$L_{\text{MSE}} = \frac{1}{N}\sum_{i=1}^{N}(\eta^l(z_i^l) - \eta^l(\hat{z}_i^l))^2 + \frac{1}{M}\sum_{i=1}^{M}(\eta^u(z_i^u) - \eta^u(\hat{z}_i^u))^2, \tag{4}$$

where $\hat{z}$ is the representation of $tx$.

The overall loss of our model can then be written as

$$L = L_{\text{CE}} + L_{\text{BCE}} + \omega(t)L_{\text{MSE}}, \tag{5}$$

where the coefficient $\omega(t)$ is a ramp-up function. This is widely used in semi-supervised learning (Laine & Aila, 2017; Tarvainen & Valpola, 2017). Following (Laine & Aila, 2017; Tarvainen & Valpola, 2017), we use the sigmoid-shaped function $\omega(t) = \lambda e^{-5(1-\frac{t}{T})^2}$, where $t$ is current time step and $T$ is the ramp-up length and $\lambda \in \mathbb{R}_+$.

## 2.5 INCREMENTAL LEARNING SCHEME

We also explore a setting analogous to incremental learning. In this approach, after tuning on the labelled set (end of section 2.1), we extend the head $\eta^l$ to $C^u$ new classes, so that $\eta^l : \mathbb{R}^d \to \mathbb{R}^{C^l+C^u}$. The head parameters for the new classes are initialized randomly. The model is then trained using the same loss eq. (5), but the cross-entropy part of the loss is evaluated on both labelled and unlabelled data $D^l$ and $D^u$. Since the cross-entropy requires labels, for the unlabelled data we use the *pseudo-labels* $\hat{y}_i^u$, which are generated on-the-fly from the head $\eta^u$ at each forward pass.

The advantage is that this approach *increments* $\eta^l$ to discriminate both old and new classes, which is often desirable in applications. It also creates a feedback loop that causes the features $z_i^u$ to be

|  | CIFAR-10 | CIFAR-100 | SVHN |
|---|---|---|---|
| Ours w/o Con | 82.6±12.0% | 61.8±3.6% | 61.3±1.9% |
| Ours w/o CE | 84.7±4.4% | 58.4±2.7% | 59.7±6.6% |
| Ours w/o BCE | 26.2±2.0% | 6.6±0.7% | 24.5±0.5% |
| Ours w/o S.S. | 89.4±1.4% | 67.4±2.0% | 72.9±5.0% |
| Ours full | **90.4±0.5%** | **73.2±2.1%** | **95.0±0.2%** |
| Ours w/ I.L. | **91.7±0.9%** | **75.2±4.2%** | **95.2±0.3%** |

Table 1: **Ablation study.** "w/o Con." means without consistency constraints; "w/o CE" means without the cross entropy loss for training on labeled data. "w/o BCE" means without binary cross entropy loss for training on unlabeled data; "w/o S.S." means without self-supervision.

refined, which in turn generates better pseudo-labels $\hat{y}_i^u$ for $D^u$ from the head $\eta^u$. In this manner, further improvements can be obtained by this cycle of positive interactions between the two heads during training.

## 3 EXPERIMENTS

### 3.1 DATA AND EXPERIMENTAL DETAILS

We evaluate our models on a variety of standard benchmark datasets: CIFAR-10 (Krizhevsky & Hinton, 2009), CIFAR-100 (Krizhevsky & Hinton, 2009), SVHN (Netzer et al., 2011), OmniGlot (Lake et al., 2015), and ImageNet (Deng et al., 2009). Following Han et al. (2019), we split these to have 5/20/5/654/30 classes respectively in the unlabelled set. In addition, for OmniGlot and ImageNet we use 20 and 3 different splits respectively, as in Han et al. (2019), and report average clustering accuracy. More details on the splits can be found in appendix A.

**Evaluation metrics.** We adopt *clustering accuracy* (ACC) to evaluate the clustering performance of our approach. The ACC is defined as follow:

$$\max_{g \in \mathrm{Sym}(L)} \frac{1}{N} \sum_{i=1}^{N} \mathbb{1}\left\{\overline{y}_i = g\left(y_i\right)\right\}, \tag{6}$$

where $\overline{y}_i$ and $y_i$ denote the ground-truth label and clustering assignment for each data point $x_i^u \in D^u$ respectively, and $\mathrm{Sym}(L)$ is the group permutations of $L$ elements (this discounts the fact that the cluster indices may not be in the same order as the ground-truth labels). Permutations are optimized using the Hungarian algorithm (Kuhn, 1955).

**Implementation details.** We use the ResNet-18 (He et al., 2016) architecture, except for OmniGlot for which we use a VGG-like network (Simonyan & Zisserman, 2015) with six layers to make our setting directly comparable to prior work. We use SGD with momentum (Sutskever et al., 2013) as optimizer for all but the OmniGlot dataset, for which we use Adam (Kingma & Ba, 2014). For all experiments we use a batch size of 128 and $k = 5$ which we found worked consistently well across datasets (see appendix D). More details about the hyper-parameters can be found in appendix B.

### 3.2 ABLATION STUDY

We validate the effectiveness of the components of our method by ablating them and measuring the resulting ACC on the unlabelled data. Note that, since the evaluation is restricted to the unlabelled data, we are solving a clustering problem. The same unlabelled data points are used for both training and testing, except that data augmentation (i.e. image transformations) is not applied when computing the cluster assignments. As can be seen in table 1, all components have a significant effect as removing any of them causes the performance to drop substantially. Among them, the BCE loss is by far the most important one, since removing it results in a dramatic drop of 40–60% absolute ACC points. For example, the full method has ACC $90.4\%$ on CIFAR-10, while removing BCE causes the ACC to drop to $26.2\%$. This shows that that our rank-based embedding comparison can indeed generate reliable pairwise pseudo labels for the BCE loss. Without consistency, cross entropy, or self-supervision, the performance drops by a more modest but still significant $7.8\%$, $5.7\%$ and $1.0\%$ absolute ACC points, respectively, for CIFAR-10. It means that the consistency term plays a role as important as the cross-entropy term by preventing the "moving target" phenomenon described in section 2.4. Finally, by incorporating the discovered classes in the classification task, we get a further boost of $1.3\%$, $2.0\%$ and $0.2\%$ points on CIFAR-10, CIFAR-100 and SVHN respectively.

Table 2: **Novel category discovery results on CIFAR-10, CIFAR-100, and SVHN.** ACC on the unlabelled set. "w/ S.S." means with self-supervised learning.

| No | | CIFAR-10 | CIFAR-100 | SVHN |
|------|-----------------------------------|---------------|---------------|---------------|
| (1) | $k$-means (MacQueen, 1967) | 65.5±0.0 % | 56.6±1.6% | 42.6%±0.0 |
| (2) | KCL (Hsu et al., 2018) | 66.5±3.9% | 14.3±1.3% | 21.4%±0.6 |
| (3) | MCL (Hsu et al., 2019) | 64.2±0.1% | 21.3±3.4% | 38.6%±10.8 |
| (4) | DTC (Han et al., 2019) | 87.5±0.3% | 56.7±1.2% | 60.9%±1.6 |
| (5) | $k$-means (MacQueen, 1967) w/ S.S. | 72.5±0.0% | 56.3±1.7% | 46.7±0.0% |
| (6) | KCL (Hsu et al., 2018) w/ S.S. | 72.3±0.2% | 42.1±1.8% | 65.6±4.9% |
| (7) | MCL (Hsu et al., 2019) w/ S.S. | 70.9±0.1% | 21.5±2.3% | 53.1±0.3% |
| (8) | DTC (Han et al., 2019) w/ S.S. | 88.7±0.3% | 67.3±1.2% | 75.7±0.4% |
| (9) | Ours | **90.4±0.5%** | **73.2±2.1%** | **95.0±0.2%** |
| (10) | Ours w/ I.L. | **91.7±0.9%** | **75.2±4.2%** | **95.2±0.2%** |

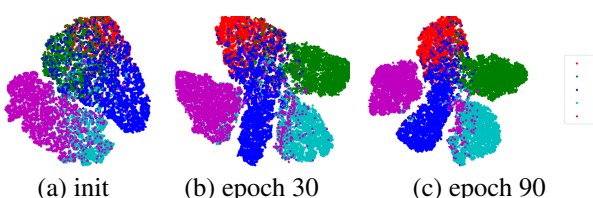

(a) init (b) epoch 30 (c) epoch 90

Figure 2: **Evolution of the t-SNE during the training of CIFAR-10.** Performed on unlabelled data (i.e., instances of dog, frog, horse, ship, truck). Colors of data points denote their ground-truth labels.

## 3.3 NOVEL CATEGORY DISCOVERY

We compare our method to baselines and state-of-the-art methods for new class discovery, starting from CIFAR-10, CIFAR-100, and SVHN in table 2. The first baseline (row 5 in table 2) amounts to applying $k$-means (MacQueen, 1967) to the features extracted by the fine-tuned model (the second step in section 2.1), for which we use the $k$-means++ (Arthur & Vassilvitskii, 2007) initialization. The second baseline (row 1 in table 2) is similar, but uses as feature extractor a model trained from scratch using only the labelled images, which corresponds to a standard transfer learning setting. By comparing rows 1, 5 and 9 in table 2, we can see that our method substantially outperforms $k$-means. Next, we compare with the KCL (Hsu et al., 2018), MCL (Hsu et al., 2019) and DTC (Han et al., 2019) methods. By comparing rows 2–4 to 9, we see that our method outperforms these by a large margin. We also try to improve KCL, MCL and DTC by using the same self-supervised initialization we adopt (section 2.1), which indeed results in an improvement (rows 2–4 vs 6–8). However, their overall performance still lags behind ours by a large margin. For example, our method of section 2.4 achieves 95.0% ACC on SVHN, while "KCL w/ S.S.", "MCL w/ S.S." and "DTC w/ S.S." achieve only 65.6%, 53.1% and 75.7% ACC, respectively. Similar trends hold for CIFAR-10 and CIFAR-100. Finally, the incremental learning scheme of section 2.5 results in further improvements, as can be seen by comparing rows 9 and 10 of table 2.

In fig. 2, we show the evolution of the learned representation on the unlabelled data on CIFAR-10 using t-SNE (van der Maaten & Hinton, 2008). As can be seen, while the clusters overlap in the beginning, they become more and more separated as the training progresses, showing that our model can effectively discover novel visual categories without labels and learn meaningful embeddings for them.

We further compare our method to others on two more challenging datasets, OmniGlot and ImageNet, in table 3. For OmniGlot, results are averaged over the 20 alphabets in the *evaluation* set (see appendix A); for ImageNet, results are averaged over the three 30-class unlabelled sets used in (Hsu et al., 2018; 2019). Since we have a relatively larger number of labelled classes in these two datasets, we follow (Han et al., 2019) and use metric learning on the labelled classes to pre-train the feature extractor, instead of the self-supervised learning. We empirically found that self-supervision does not provide obvious gains for these two datasets. This is reasonable since the data in the labelled sets of these two datasets are rather diverse and abundant, so metric learning can provide good feature initialization as there is less class-specific bias due to the large number of pre-training classes. However, by comparing rows 1 and 5 in table 3, it is clear that metric learning alone is not sufficient for the task of novel category discovery. Our method substantially outperforms the $k$-means results obtained using the features from metric learning — by 11.9% and 10.6% on OmniGlot and ImageNet respectively. Our method also substantially outperforms the current state-of-the-art, achieving 89.1%

| No | | OmniGlot | ImageNet |
|---|---|---|---|
| (1) | $k$-means (MacQueen, 1967) | 77.2% | 71.9% |
| (2) | KCL (Hsu et al., 2018) | 82.4% | 73.8% |
| (3) | MCL (Hsu et al., 2019) | 83.3% | 74.4% |
| (4) | DTC (Han et al., 2019) | 89.0% | 78.3% |
| (5) | **Ours** | **89.1%** | **82.5%** |

Table 3: **Novel category discovery results on OmniGlot and ImageNet.** ACC on the unlabelled set.

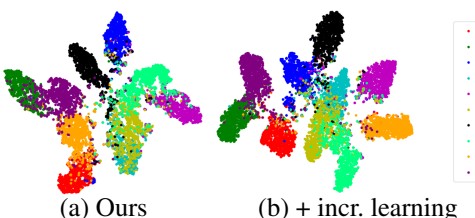

(a) Ours      (b) + incr. learning

Figure 3: **t-SNE on CIFAR-10: impact of incremental Learning.** (a) representation on the labelled and (b) unlabelled CIFAR classes. Colors of data points denote their ground-truth labels. We observe a bigger overlap in (a) between the "old" class 3 and the "new" class 5 when not incorporating Incremental Learning.

and 82.5% ACC on OmniGlot and ImageNet respectively, compared with 89.0% and 78.8% of (Han et al., 2019), thus setting the new state-of-the-art.

## 3.4 INCREMENTAL LEARNING

Table 4: **Incremental Learning with the novel categories.** "old" refers to the ACC on the labelled classes while "new" refers to the unlabelled classes in the *testing set*. "all" indicates the whole testing set. It should be noted that the predictions are not restricted to their respective subset. Standard deviation can be found in appendix C.

| | CIFAR-10 | | | CIFAR-100 | | | SVHN | | |
|---|---|---|---|---|---|---|---|---|---|
| Classes | old | new | all | old | new | all | old | new | all |
| KCL w/ S.S. | 79.4% | 60.1% | 69.8% | 23.4% | 29.4% | 24.6% | 90.3% | 65.0% | 81.0% |
| MCL w/ S.S. | 81.4% | 64.8% | 73.1% | 18.2% | 18.0% | 18.2% | 94.0% | 48.6% | 77.2% |
| DTC w/ S.S. | 58.7% | 78.6% | 68.7% | 47.6% | 49.1% | 47.9% | 90.5% | 72.8% | 84.0% |
| Ours w/ I.L. | **90.6%** | **88.8%** | **89.7%** | **71.2%** | **56.8%** | **68.3%** | **96.3%** | **96.1%** | **96.2%** |

Here, we further evaluate our incremental scheme for novel category discovery as described in section 2.5. Methods for novel category discovery such as (Han et al., 2019; Hsu et al., 2019; 2018) focus on obtaining the highest clustering accuracy for the new unlabelled classes, but may forget the existing labelled classes in the process. In practice, forgetting is not desirable as the model should be able to recognize both old and new classes. Thus, we argue that the classification accuracy on the labelled classes should be assessed as well, as for any incremental learning setting. Note however that our setup differs substantially from standard incremental learning (Rebuffi et al., 2017; Lopez-Paz & Ranzato, 2017; Shmelkov et al., 2017; Aljundi et al., 2018) where every class is labelled and the focus is on using limited memory. In our case, we can store and access the original data without memory constraints, but the new classes are unlabelled, which is often encountered in applications.

By construction (section 2.5), our method learns the new classes on top of the old ones incrementally, out of the box. In order to compare to methods such as KCL, MCL and DTC that do not have this property, we proceed as follows. First, the method runs as usual to cluster the unlabelled portion of the data, thus obtaining pseudo-labels for it, and learning a feature extractor as a byproduct. Then, the feature extractor is used to compute features for both the labelled and unlabelled training data, and a linear classifier is trained using labels and pseudo-labels, jointly on all the classes, old and new.

We report in table 4 the performance of the resulting joint classifier networks *on the testing set* of each dataset (this is now entirely disjoint from the training set). Our method has similar performances on the old and new classes for CIFAR-10 and SVHN, as might be expected as the split between old and new classes is balanced. In comparison, the feature extractor learned by KCL and MCL works much better for the old classes (e.g., the accuracy discrepancy between old and new classes is 25.3% for KCL on SVHN). Conversely, DTC learns features that work better for the new classes, as shown by the poor performance for the old classes on CIFAR-10. Thus, KCL, MCL and DTC learn representations that are biased to either the old or new classes, resulting overall in suboptimal performance. In contrast, our method works well on both old and new classes; furthermore, it drastically outperforms existing methods on both.

## 4  RELATED WORK

Our work draws inspiration from semi-supervised learning, transfer learning, clustering, and zero-shot learning. We review below the most relevant contributions.

In semi-supervised learning (SSL) (Chapelle et al., 2006), a partially labelled training dataset is given and the objective is to learn a model that can propagate the labels from the labelled data to unlabelled data. Most SSL methods focus on the classification task where, usually both labelled and unlabelled points belong to the same set of classes. On the contrary, our goal is to handle the case where the unlabelled data classes differ from the labelled data. Oliver et al. (2018) summarizes the state-of-the-art SSL methods. Among them, the consistency-based methods appeared to be the most effective. Rasmus et al. (2015) propose a ladder network which is trained on both labelled and unlabelled data using a reconstruction loss. Laine & Aila (2017) simplifies this ladder network by enforcing prediction consistency between a data point and its augmented counterpart. As an alternative to data augmentation, they also consider a regularization method based on the exponential moving average (EMA) of the predictions. This idea is further improved by Tarvainen & Valpola (2017): instead of using the EMA of predictions, they propose to maintain the EMA of model parameters. The consistency is then measured between the predictions of the current model (student) and the predictions of the EMA model (teacher). More recently (and closer to our work) practitioners have also combined SSL with self-supervision(Rebuffi et al., 2019; Zhai et al., 2019) to leverage dataset with very few annotations.

Transfer learning (Pan & Yang, 2010; Weiss et al., 2016; Tan et al., 2018) is an effective way to reduce the amount of data annotations required to train a model by pre-training the model on an existing dataset. In image classification, for example, it is customary to start from a model pre-trained on the ImageNet (Deng et al., 2009) dataset. In most transfer learning settings, however, both the source data and the target data are fully annotated. In contrast, our goal is to transfer information from a labelled dataset to an unlabelled one.

Many classic (e.g., Aggarwal & Reddy (2013); MacQueen (1967); Comaniciu & Meer (1979); Ng et al. (2001)) and deep learning (e.g., Xie et al. (2016); Chang et al. (2017); Dizaji et al. (2017); Yang et al. (2017; 2016); Hsu et al. (2018; 2019)) clustering methods have been proposed to automatically partition an unlabelled data collection into different classes. However, this task is usually ill-posed as there are multiple, equally valid criteria to partition most datasets. We address this challenge by learning the appropriate criterion by using a labelled dataset, narrowing down what constitutes a proper class. We call this setting "transfer clustering".

To the best of our knowledge, the work most related to ours are (Hsu et al., 2018; 2019; Han et al., 2019). Han et al. (2019) also consider discovering new classes as a transfer clustering problem. They first learn a data embedding by using metric learning on the labelled data, and then fine-tune the embedding and learn the cluster assignments on the unlabelled data. In (Hsu et al., 2018; 2019), the authors introduce KCL and MCL clustering methods. In both, a similarity prediction network (SPN), also used in (Hsu et al., 2016), is first trained on a labelled dataset. Afterwards, the pre-trained SPN is used to provide binary pseudo labels for training the main model on an unlabelled dataset. The overall pipelines of the two methods are similar, but the losses differ: KCL uses a Kullback-Leibler divergence based contrastive loss equivalent to the BCE used in this paper (eq. (3)), and MCL uses the Meta Classification Likelihood loss. Zero-shot learning (ZSL) (Xian et al., 2018; Fu et al., 2018) can also be used to recognize new classes. However, differently from our work, ZSL also requires additional side information (e.g., class attributes) in addition to the raw images.

Finally, other works (Dean et al., 2013; Yagnik et al., 2011) discuss the application of rank statistics to measuring the similarity of vectors; however, to the best of our knowledge, we are the first to apply rank statistics to the task of novel category discovery using deep neural networks.

## 5  CONCLUSIONS

In this paper, we have looked at the problem of discovering new classes in an image collection, leveraging labels available for other, known classes. We have shown that this task can be addressed very successfully by a few new ideas. First, the use of self-supervised learning for bootstrapping the image representation trades off the representation quality with its generality, and for our problem this leads to a better solution overall. Second, we have shown that rank statistics are an effective method to compare noisy image descriptors, resulting in robust data clustering. Third, we have

shown that jointly optimizing both labelled recognition and unlabelled clustering in an incremental learning setup can reinforce the two tasks while avoiding forgetting. On standard benchmarks, the combination of these ideas results in much better performance than existing methods that solve the same task. Finally, for larger datasets with more classes and diverse data (e.g., ImageNet) we note that self-supervision can be bypassed as the pretraining on labelled data already provides a powerful enough representation. In such cases, we still show that the rank statistics for clustering gives drastic improvement over existing methods.

## 6  Acknowledgments

This work is supported by the EPSRC Programme Grant Seebibyte EP/M013774/1, Mathworks/DTA DFR02620, and ERC IDIU-638009.

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

## A    DATASET SPLITS

For CIFAR-10 and SVHN we keep the labels of the five first categories (namely airplane, automobile, bird, cat, deer for CIFAR-10, 0–4 for SVHN) and keep the rest of the data as the unlabelled set. For CIFAR-100 we use the first 80 categories as labelled data while the rest are used for the unlabelled set. Following Hsu et al. (2018; 2019), for OmniGlot, each of the 20 alphabets in *evaluation* set (with 20–47 categories, 659 characters/class) is used as unlabelled data, and all the 30 alphabets in *background* set are used as labelled set (964 characters/class). For ImageNet, we follow Hsu et al. (2018; 2019) to use the 882/118 classes split proposed in Vinyals et al. (2016), and use the three 30-class subsets sampled from the 118 classes as unlabelled sets.

## B    IMPLEMENTATION DETAILS

In the first self-supervised training step, otherwise mentioned, we trained our model with the pretext task of rotation predictions (i.e., a four-class classification: $0°$, $90°$, $180°$, and $270°$) for 200 epochs and a step-wise decaying learning rate starting from 0.1 and divided by 5 at epochs 60, 120, and 160.

In the second step of our framework (i.e., supervised training using labelled data), we fine-tuned our model on the labelled set for 100 epochs and a step-wise decaying learning rate starting from 0.1 and halved every 10 epochs. From this step onward we fix the first three convolutional blocks of the model, and fine-tuned the last convolutional block together with the linear classifier.

Finally, in the last joint training step, we fine-tuned our model for 200/100/90 epochs for {CIFAR-10, CIFAR-100, SVHN}/OmniGlot/ImageNet, which was randomly sampled from the merged set of both labelled and unlabelled data. The initial learning rate was set to 0.1 for all datasets, and was decayed with a factor of 10 at the 170th/{30th, 60th} epoch for {CIFAR-10, CIFAR-100, SVHN}/ImageNet. The learning rate of 0.01 was kept fixed for OmniGlot. For the consistency regularization term, we used the ramp-up function as described in section 2.4 with $\lambda = \{5.0, 50.0, 50.0, 100.0, 10.0\}$, and $T = \{50, 150, 80, 1, 50\}$ for CIFAR-10, CIFAR-100, SVHN, OmniGlot, and ImageNet respectively.

In the incremental learning setting, all previous hyper parameters remain the same for our method. We only add a ramp-up on the cross entropy loss on unlabelled data. The ramp-up length is the same as the one used for eq. (4) and we use for all experiments a coefficient of 0.05. For all other methods we trained the classifier for 150 epochs with SGD with momentum and learning rate of 0.1 divided by 10 at epoch 50.

We implemented our method using PyTorch 1.1.0 and ran experiments on NVIDIA Tesla M40 GPUs. Following (Han et al., 2019), our results were averaged over 10 runs for all datasets, except for ImageNet which was averaged over the three 30-class subsets. In general, we found the results were stable. Our code is publicly available at `http://www.robots.ox.ac.uk/~vgg/research/auto_novel`.

## C    STANDARD DEVIATION OF INCREMENTAL LEARNING EXPERIMENT IN TABLE 4

Table 5: **Incremental Learning with the novel categories.** "old" refers to the standard deviation ACC on the labelled classes while "new" refers to the unlabelled classes in the *testing set*. "all" indicates the whole testing set. It should be noted that the predictions are not restricted to their respective subset.

| Classes | CIFAR-10 | | | CIFAR-100 | | | SVHN | | |
|---|---|---|---|---|---|---|---|---|---|
| | old | new | all | old | new | all | old | new | all |
| KCL w/ S.S. | 0.6% | 0.6% | 0.1% | 0.3% | 0.3% | 0.2% | 0.3% | 0.5% | 0.1% |
| MCL w/ S.S. | 0.4% | 0.4% | 0.1% | 0.3% | 0.1% | 0.2% | 0.2% | 0.3% | 0.1% |
| DTC w/ S.S. | 0.6% | 0.2% | 0.3% | 0.2% | 0.2% | 0.2% | 0.3% | 0.2% | 0.1% |
| Ours w/ I.L. | 0.2% | 0.2% | 0.1% | 0.1% | 0.3% | 0.1% | 0.1% | 0.0% | 0.1% |

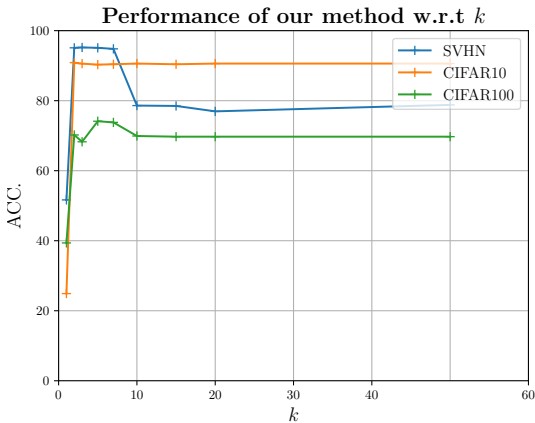

Figure 4: Performance evolution with respect to $k$. We report results for $k = \{1, 2, 3, 5, 7, 10, 15, 20, 50\}$.

## D   IMPACT OF $k$ OVER RESULTS

We provide an additional study of the evolution of performances of our method with respect to $k$. We results on SVHN/CIFAR10/CIFAR100 in fig. 4. We found that $k = \{5, 7\}$ gave the best results overall. We also found that for all values of $k$ except 1 results were in general stable.

## E   RESULTS WITH AN UNKNOWN NUMBER OF CLASSES

While in our work we assume the number of new classes $C^u$ to be known a priori, this hypothesis can be restrictive in practice. Instead, one can estimate the number of classes in $D^u$ using recent methods such as DTC (Han et al., 2019). In table 6 we compare ACC of KCL (Hsu et al., 2018), KCL (Hsu et al., 2019), DTC(Han et al., 2019) and our method on unlabelled splits of OmniGlot and ImageNet datasets with $C^u$ computed from DTC. We note that our method again reaches the state-of-the-art on ImageNet and is on par with the state-of-the-art on OmniGlot.

Table 6: **Novel category discovery results with unknown** $C^u$.

| No | | OmniGlot | ImageNet |
|----|-----------------------|----------|----------|
| (1) | KCL (Hsu et al., 2018) | 80.3% | 71.4% |
| (2) | MCL (Hsu et al., 2019) | 80.5% | 72.9% |
| (3) | DTC (Han et al., 2019) | **87.0**% | 77.6% |
| (4) | Ours | 85.4% | **80.5**% |

