# OpenReview forum: "Automatically Discovering and Learning New Visual Categories with Ranking Statistics"
_ICLR.cc/2020/Conference — Accept (Poster)_

### Official Review · AnonReviewer1 · 2019-10-23
**Official Blind Review #1**

**Rating:** 3

**Review:**



  This paper tackles the problem of unsupervised object discovery, whereby a labeled dataset must be leveraged in order to then cluster an unlabeled dataset with a set of unknown categories. The paper contributes three main ideas to succeed at this task, namely 1) use of self-supervised learning to initialize the representations in a way that doesn't bias them to the labeled data, 2) a robust rank-based metric to generate estimates of similarity/dissimilarity along with consistency-based regularization to improve optimization, and 3) Joint optimization/refinement using a combination of labeled/unlabeled losses, as well as ability to learn incrementally without forgeting the original labeled classes. Results are shown on a range of datasets including OmniGlot, ImageNet, CIFAR-10, CIFAR-100, and SVHN. The results demonstrate improvement over the current state of art for this task.

  Overall, my current vote for this paper is a weak reject. The main reason is that the paper really combines a set of known methods (self-supervised learning, consistency-based regularization, and an ad-hoc training regimen. On the other hand, the paper is well-written and provides nice rigorous experiments showing clear improvements over state of art by porting these known techniques from different domains (self/semi-supervised learning). However, if satisfactory answers to questions below are given, I am willing to change my rating.

Main Argument

Strengths

  - Overall the paper tackles an interesting problem, and does so in a way that achieves good results beyond state of art.

  - An ablation study is provided which shows the contribution of different parts of the method.

  - The paper is very well-motivated, written, and methods are described nicely and succinctly.

Weaknesses

  - Clearly the methods employed are, by themselves, not novel and have been used for a range of other ML problem formulations. What is the clear contribution/novelty of the work?

  - While the paper raises an interesting motivation about not biasing feature learning by using self-supervised learning, it's not clear to me that this claim is justified. What is the evidence for this, besides better performance? While the ablation w/o self-supervised learning performs more poorly than everything combined, the other parts of the ablation (with self-supervised learning) also perform poorly. Clearly, there is some interaction between the different aspects of the method, but I am not sure what that is. Why is the full combination so much better than if any one thing is removed?

  - The method is very similar to KCL (e.g. loss (3) is the same and justified via a graphical model formulation in that paper). The paper does not really read like it is building off of that though, which seems a bit misleading. Do the authors believe there is additional novelty, or is it a matter of adding the three contributions to KCL? What is the major reason that self-supervision with KCL still doesn't do as well? As far as I can tell, the only difference is self-supervised learning, the fact that you have a manual 3-stage curriculum, and the robust ranking method/consistency loss (i.e. the three contributions).

  - You assume that the number of clusters is known; this is one of the advantages of all of the prior work in that they can estimate this. How well does the method work if the number of clusters is not known?

Additional comments not related to final vote:

  - The paper seems to start out in a way that implies the problem is new; citations should be provided to the three main compared works in the intro to make it clear that this is a follow-on to an existing problem. Further, as mentioned above MCL/KCL (especially the latter) are very similar in nature and introduced this problem (including [1] which also used unsupervised feature learning in one condition). These should be included as "work most related to ours" given that they came before Han et al.

  [1] Deep Image Category Discovery using a Transferred Similarity Function, https://arxiv.org/abs/1612.01253.



**Experience Assessment:**

I have published in this field for several years.

**Review Assessment: Checking Correctness Of Derivations And Theory:**

N/A

**Review Assessment: Checking Correctness Of Experiments:**

I carefully checked the experiments.

**Review Assessment: Thoroughness In Paper Reading:**

N/A

---

> ### Author Response · Authors · 2019-11-10
> **Answer to Reviewer 1 (1/2)**
>
> We thank the reviewer for the insightful comments and suggestions.
>
> > Clearly the methods employed are, by themselves, not novel and have been used for a range of other ML problem formulations. What is the clear contribution/novelty of the work?
>
> The main novelty of this work is to demonstrate the effectiveness of rank statistics in transferring the knowledge contained in a labelled dataset to the problem of clustering an unlabelled dataset. Empirically, we demonstrate that, due to this innovation, we can outperform existing methods by a large margin. Meanwhile, to the best of our knowledge, we are the first to use rank statistics to generate pseudo labels on-the-fly for novel category discovery.
>
> The second significant novelty is the introduction of the two classification heads on top of features computed by the same network trunk. One head is used to learn new classes on the unlabelled data, while the other head maintains a meaningful representation for the labelled subset of the data. We show that this second head improves classification accuracy on the unlabelled set (see Table 1 second row), which is non-trivial.
>
> Finally, as far as we know, we are also the first to combine incremental learning with new class discovery. We show in table 2 and 4 that incremental learning was beneficial to new class discovery, particularly on the test set (Table 4).
>
> > While the paper raises an interesting motivation about not biasing feature learning by using self-supervised learning, it's not clear to me that this claim is justified. What is the evidence for this, besides better performance? While the ablation w/o self-supervised learning performs more poorly than everything combined, the other parts of the ablation (with self-supervised learning) also perform poorly. Clearly, there is some interaction between the different aspects of the method, but I am not sure what that is. Why is the full combination so much better than if any one thing is removed?
>
> We introduce self-supervision as a means to limit overfitting during the first phase of the algorithm: here in fact the deep neural network features are learned using the labelled data, and we do not wish to over-specialise the features to just recognise those labels (a problem that affects other methods). Since this is a general strategy that can improve other methods as well, we did test it throughout the board, demonstrating its generality and effectiveness, and giving the other methods a better chance to compete against our other improvements.
>
> As for the other aspects of the method:
>
>     _The aim of the BCE loss is to cluster the unlabelled data to help discovering the new classes. Our method cannot run without the BCE loss as the other parts of the method do not enforce explicit classification constraints on the unlabelled data. This was verified in Table 1: results without BCE dropped to the level of chance on all three datasets.
>
>     _ If we remove the CE loss, the method is unable to keep learning from the labelled images and their ground-truth class labels. Therefore, the ability of the method to discover and classify the new classes drops significantly by 35.3%/5.7%/14.8% on respectively SVHN/CIFAR10/CIFAR100. This is consistent with the key motivation of our work, which is to use the classification on labelled classes to learn how to label the unknown classes.
>
> _ By removing consistency, we no longer enforce the cluster prediction to be consistent to different data transformations.
>
> Note that the different losses capture independent cues/signals that help the method to learn better. As we show in the ablation, removing any of them results in a reduced performance.

---

> > ### Author Response · Authors · 2019-11-10
> > **Answer to Reviewer 1 (2/2)**
> >
> > >The method is very similar to KCL (e.g. loss (3) is the same and justified via a graphical model formulation in that paper). The paper does not really read like it is building off of that though, which seems a bit misleading. Do the authors believe there is additional novelty, or is it a matter of adding the three contributions to KCL? What is the major reason that self-supervision with KCL still doesn't do as well? As far as I can tell, the only difference is self-supervised learning, the fact that you have a manual 3-stage curriculum, and the robust ranking method/consistency loss (i.e. the three contributions).
> >
> > There are three key differences:
> >
> > We agree that loss (3) is similar to KCL, but the way the pseudo labels are obtained is entirely different and a key advantage of our approach.
> > Table 2 shows that self-supervision improves KCL and MCL too (rows 2-3 vs rows 6-7). However, our ranking-based pseudo labels evolve over time, while KCL and MCL use fixed pseudo labels, which is why our method still outperforms them even after self-supervision is backported to KCL and MCL.
> > We also show that our method without self-supervision already outperforms KCL (row 4 in Table 1 vs row 2 and row 6 in Table 2) due to the rank statistics we use to propose pseudo-labels.
> >
> > We think that, for these three reasons, there is non-negligible novelty compared to KCL.
> >
> > > You assume that the number of clusters is known; this is one of the advantages of all of the prior work in that they can estimate this. How well does the method work if the number of clusters is not known?
> >
> > This is included in the response to R3, and we copy the text here for the convenience of the reviewer:
> >
> > To the best of our knowledge, most of the related methods (MCL, KCL, DTC) also assume to know the number of classes during training. When the number of classes is unknown, MCL and KCL simply set it to a nominal large value.
> >
> > However, the recent paper by (Han et al ICCV 19) introduced a method to estimate the number of novel classes automatically. Therefore, we have assessed our method on OmniGlot and ImageNet by using the number of classes estimated by (Han et al ICCV 19), comparing our clustering results with the ones obtained of MCL, KCL, and DTC for the same number of classes.
> >
> > By doing so, on OmniGlot, we obtained 85.4% accuracy compared to 80.3/80.5/87.0 obtained by KCL/MCL/DTC. Hence, in this case our method performs better than KCL and MCL, and close to the state-of-the-art result obtained by DTC. Meanwhile, please note that when the number of classes are known, our method performs better than other methods by a large margin (see Table 2, 3, 4 in the paper).
> > Results on ImageNet are currently running and we will update the tables as soon as they are ready.
> >
> > > The paper seems to start out in a way that implies the problem is new; citations should be provided to the three main compared works in the intro to make it clear that this is a follow-on to an existing problem. Further, as mentioned above MCL/KCL (especially the latter) are very similar in nature and introduced this problem (including [1] which also used unsupervised feature learning in one condition). These should be included as "work most related to ours" given that they came before Han et al.
> >
> > We thank the reviewer for this suggestion, we have changed the introduction and related work to better reflect the points raised in this comment.

---

### Official Review · AnonReviewer2 · 2019-10-24
**Official Blind Review #2**

**Rating:** 6

**Review:**

Summary:
This paper addresses the problem of clustering unseen classes. To learn a robust feature extractor, this paper proposes a multi-stage training framework, which leverages different supervised manners in each stage. Specifically, they initialize the network using the self-supervised learning on the union of all available data and then further finetune it using labelled data. Based on this, they propose the rank statistics which leverages the activation knowledge on labelled classes and rank the activated dimensions. Unseen data having similar rank results are clustered to obtain the initial pseudo labels. Finally, the network is jointly optimized with the ground-truth and generated pseudo labels (the pseudo ones will be updated during training). Extensive experiments on 5 datasets show that their method has significant advantages over SOTA owing to the learned robust feature extractor.

+Strengths:
1. The writing of this paper is satisfactory. Both the related works, motivations and technical details are clearly introduced.
2. The experiments are solid. They evaluate their method on five popular object datasets and both ablation of each components and comparison with SOTA are shown in the paper.
3. This paper also shows that their method has good ability of avoiding forgetting of old (seen) classes, which may provide some insights about feature extraction for improving incremental learning.

-Weaknesses:
1. Except for the experimental evaluation, what is the advantage of rank statistics over directly comparing feature vectors? Why robust?
2. Some experimental issues. a) how will the choice of k in top-k rank influence the performance? b) why the advantages of incorporating incremental learning on SVHN and CIFAR-10 are not obvious? c) why evaluation of incremental learning on CIFAR-100 is not well (acc difference between old and new is larger than other datasets). Besides, what is acc. performance of old classes with only labeled data for training.
3. Typos. In Sec.4, the writing of KCL (KLC) and MCL (MLC) is not consistent.

**Experience Assessment:**

I have published one or two papers in this area.

**Review Assessment: Checking Correctness Of Derivations And Theory:**

I carefully checked the derivations and theory.

**Review Assessment: Checking Correctness Of Experiments:**

I carefully checked the experiments.

**Review Assessment: Thoroughness In Paper Reading:**

I read the paper thoroughly.

---

> ### Author Response · Authors · 2019-11-10
> **Answer to Reviewer 2**
>
> We thank the reviewer for helpful comments and suggestions.
>
> >1. Except for the experimental evaluation, what is the advantage of rank statistics over directly comparing feature vectors? Why robust?
>
> When directly comparing feature vectors, if we replace rank statistics comparison with the cosine similarity between feature vectors, the performance dropped to 39.8/11.4/63.5 on CIFAR 10/CIFAR100/SVHN (vs ours 91.7/ 75.2/95.2).
>
> As discussed in [1,2,3], rank statistics are robust to numerical perturbations while giving a good indication of inherent agreement between vectors. As shown in [2], rank statistics are especially helpful when the dimension of the vectors is large, as it is in our case, effectively reducing the effects of the “curse of dimensionality” when comparing vectors via the standard distances.
>
> [1] J. Yagnik, D.Strelow, D. A. Ross, R. Lin. The power of comparative reasoning, In ICCV 2011
> [2] J. Friedman. An Overview of Predictive Learning and Function Ap- proximation. In From Statistics to Neural Networks’94
> [3] P. Diaconis, R. Graham. Spearman’s footrule as a measure of disarray. In J. Roy. Statistical Society’77
>
> >2.a) how will the choice of k in top-k rank influence the performance?
> This is included in the response to R3, and we copy the text here for the convenience of the reviewer:
>
> Following this suggestion, we have added a new figure (Figure 4 in the appendix) to study how performance depends on the value of k (the text in section 3.1 has also been slightly adjusted to refer to this figure). We found that k=5 was performing consistently well across datasets, with k=7 obtaining similar results. In general, different choices of k did not affect much the results. However, when k is extremely small (e.g., k=1), the performance drops drastically, as the strongest activation in the last layer of the network is in this case unlikely to be discriminative enough to compare samples.
>
> > 2.b) why the advantages of incorporating incremental learning on SVHN and CIFAR-10 are not obvious?
>
> We agree that the improvement on SVHN and CIFAR10 in Table 2 is not very large. We hypothesize that this is because the accuracies on both of these datasets are already high (95.0% and 90.4% acc) without I/L, so there is little margin for improvement. Nevertheless, there are still some gains (+0.2% and +1.3% on SVHN and CIFAR10), which shows that incorporating incremental learning still helps.
>
> > 2.c) why evaluation of incremental learning on CIFAR-100 is not well (acc difference between old and new is larger than other datasets).
>
> CIFAR100 is just much more challenging than the other two datasets due to the much smaller number of images per class and the much larger number of classes. In particular, this is likely to increase the confusion between old and new classes, which explains the reduced performance.
>
>  > What is acc. performance of old classes with only labeled data for training ?
>
> Following this suggestion, we report performance on such cases. When using only labelled data for training classification, accuracies on the old classes on SVHN/CIFAR10/CIFAR100 are 97.9/96.0/74.2 respectively. While these results are an upper bound of the ones obtained in Table 4, we note that our method was the only one to reach similar performances (96.3/90.6/71.2).
>
> > 3. Typos. In Sec.4, the writing of KCL (KLC) and MCL (MLC) is not consistent.
>
> Thank you, these typos have been corrected in the updated version.

---

### Official Review · AnonReviewer3 · 2019-11-04
**Official Blind Review #3**

**Rating:** 6

**Review:**

The authors propose a methodology to discover new categories in an unlabeled dataset with the help of a label one. The authors propose the following methodology. First bootstrap some features using self-supervised learning on labeled and unlabeled data. Then transferring the knowledge of the labeled data to the unlabeled one by supposing that the representations of both are similar, the similarity being a rank statistic. Then using this knowledge a joint supervised-unsupervised objective.

Q1. The paper is about discovering new visual classes. Section 2 mention that the number of classes C^u must be known "a priori". How do you tackle this limitation? Is there any heuristic, similar to the one found in the clustering literature, that could help?

Q2. Have other losses been investigated for the clustering head? Such as the triplet loss, or deep clustering loss rather than BCE?

I would suggest to report standard deviations as the experiments were repeated 10 times on random train-test splits.

I propose a weak accept. The paper is well written, the methodology is original, the experiments are convincing and the authors will release publicly the code. My main concern is Q1, which is eluded. Second, it would be nice to have an experiment illustrating the impact of the rank k when transferring knowledge. Is k=5 always good? If not, what could influence the best value?


**Experience Assessment:**

I have read many papers in this area.

**Review Assessment: Checking Correctness Of Derivations And Theory:**

N/A

**Review Assessment: Checking Correctness Of Experiments:**

I assessed the sensibility of the experiments.

**Review Assessment: Thoroughness In Paper Reading:**

I read the paper at least twice and used my best judgement in assessing the paper.

---

> ### Author Response · Authors · 2019-11-10
> **Answer to Reviewer 3**
>
> We would like to thank the reviewer for his comments and helpful suggestions.
>
> >Q1. The paper is about discovering new visual classes. Section 2 mention that the number of classes C^u must be known "a priori". How do you tackle this limitation? Is there any heuristic, similar to the one found in the clustering literature, that could help?
>
> To the best of our knowledge, most of the related methods (MCL, KCL, DTC) also assume to know the number of classes during training. When the number of classes is unknown, MCL and KCL simply set it to a nominal large value.
>
> However, the recent paper by (Han et al ICCV 19) introduced a method to estimate the number of novel classes automatically. Therefore, we have assessed our method on OmniGlot and ImageNet by using the number of classes estimated by (Han et al ICCV 19), comparing our clustering results with the ones obtained of MCL, KCL, and DTC for the same number of classes.
>
> By doing so, on OmniGlot, we obtained 85.4% accuracy compared to 80.3/80.5/87.0 obtained by KCL/MCL/DTC. Hence, in this case our method performs better than KCL and MCL, and close to the state-of-the-art result obtained by DTC. Meanwhile, please note that when the number of classes are known, our method performs better than other methods by a large margin (see Table 2, 3, 4 in the paper).
> Results on ImageNet are currently running and we will update the tables as soon as they are ready.
>
> >Q2. Have other losses been investigated for the clustering head? Such as the triplet loss, or deep clustering loss rather than BCE?
>
> We also experimented with the triplet loss instead of BCE and found that on CIFAR10, CIFAR100 and SVHN the final results are not as good as the ones obtained by using BCE (81.2/47.0/78.0 vs 91.7/ 75.2/95.2).
>
> > I would suggest to report standard deviations as the experiments were repeated 10 times on random train-test splits.
>
> We have added standard deviations as requested (see the updated Tables 1 and 2 in the updated paper and the extended Table 4 to Table 5 in the updated appendix). In order to do this, we have rerun all the experiments 10 times with different initializations, following the standard protocol for these datasets. Results are very stable: the standard deviations on respectively SVHN/CIFAR10/CIFAR100 are 0.2%/0.5%/2.1% (Table 2 row 9) which are in line with the results we obtained with DTC (0.4%/0.3%/1.2%), MCL (0.3%/0.1%/2.3%), KCL (4.9%/0.2%/1.8%) and k-means (0.%/0.%/1.6%) when used in combination with self-supervision (Table 2 rows 5-8).
>
> >Second, it would be nice to have an experiment illustrating the impact of the rank k when transferring knowledge. Is k=5 always good? If not, what could influence the best value?
>
> Following this suggestion, we have added a new figure (Figure 4 in the appendix) to study how performance depends on the value of k (the text in section 3.1 has also been slightly adjusted to refer to this figure). We found that k=5 was performing consistently well across datasets, with k=7 obtaining similar results. In general, different choices of k did not affect much the results. However, when k is extremely small (e.g., k=1), the performance drops drastically, as the strongest activation in the last layer of the network is in this case unlikely to be discriminative enough to compare samples.

---

### Author Response · Authors · 2019-11-14
**Additional ImageNet Experiment**

In addition to our previous response to R1-Q4 and R3-Q1 w.r.t the case of unknown number of classes, we would like to inform the reviewers that the ImageNet experiment using the number of clusters computed from DTC is now completed.
Our method reached final performance of 80.5% acc outperforming DTC (77.6%), MCL (72.9%) and KCL (71.4%) methods.
Note that for fair comparison with MCL and KCL, we also reported their performance using the number of clusters computed from DTC (Table 6 in (Han et al. 2019)).

---

### Decision · Program_Chairs · 2019-12-19

**Decision:**

Accept (Poster)

**Comment:**

The paper defines a methodology to discover unknown classes in a semi-supervised learning setting, based on: i) defining a proper representation based on self-supervision on all samples; ii) defining equivalence classes on the unlabelled samples, based on ranking statistics; iii) training supervised heads aimed to predict the labels (when available) and the equivalence class indices (when unlabelled).

All reviewers agree that the ranking statistics-based heuristics is a quite innovative element of the paper. The extensive and careful experimental validation, with the ablation studies, establishes the merits of all ingredients.

Therefore, I propose acceptance of this paper.